# Diffusion Policy Attacker: Crafting Adversarial Attacks for Diffusion-based Policies

**Yipu Chen\***
Georgia Institute of Technology
ychen3302@gatech.edu

**Haotian Xue\***
Georgia Institute of Technology
htxue.ai@gatech.edu

**Yongxin Chen**
Georgia Institute of Technology
yongchen@gatech.edu

## Abstract

Diffusion models (DMs) have emerged as a promising approach for behavior cloning (BC). Diffusion policies (DP) based on DMs have elevated BC performance to new heights, demonstrating robust efficacy across diverse tasks, coupled with their inherent flexibility and ease of implementation. Despite the increasing adoption of DP as a foundation for policy generation, the critical issue of safety remains largely unexplored. While previous attack attempts have targeted deep policy networks, DP used diffusion models as the policy network, making it ineffective to be attacked using previous methods because of its chained structure and randomness injected. In this paper, we undertake a comprehensive examination of DP safety concerns by introducing adversarial scenarios, encompassing offline and online attacks, and global and patch-based attacks. We propose DP-Attacker, a suite of algorithms that can craft effective adversarial attacks across all aforementioned scenarios. We conduct attacks on pre-trained diffusion policies across various manipulation tasks. Through extensive experiments, we demonstrate that DP-Attacker has the capability to significantly decrease the performance of DP for all scenarios. Particularly in offline scenarios, DP-Attacker can generate highly transferable perturbations applicable to all frames. Furthermore, we illustrate the creation of adversarial physical patches that, when applied to the environment, effectively deceive the model. Video results are put in: https://sites.google.com/view/diffusion-policy-attacker.

## 1 Introduction

Behavior Cloning (BC) [40] is a pivotal area in robot learning: given an expert demonstration dataset, it aims to train a policy network in a supervised approach. Recently, diffusion models [16, 49] have become dominant in BC, primarily due to their strong capability in modeling multi-modal distribution. The resulting policy learner, termed Diffusion Policy (DP) [9, 18], can generate the action trajectory from a pure Gaussian noise conditioned on the input image(s). An increasing number of works are adopting DP as an action decoder for BC across various domains such as robot manipulation [12, 58, 7], long-horizon planning [35, 26] and autonomous driving [29].

Adversarial attack [31, 14] has been haunting deep neural networks (DNN) for a long time: a small perturbation on the input image will fool the DNN into making wrong decisions. Despite the remarkable success of diffusion policies in BC, their robustness under adversarial attacks [31, 14]

---

\* indicates equal contribution. Correspondence to: ychen3302@gatech.edu

38th Conference on Neural Information Processing Systems (NeurIPS 2024).

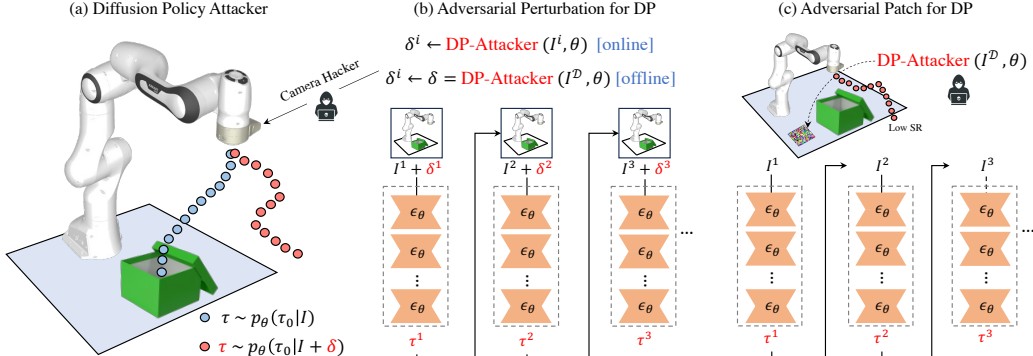

Figure 1: **Adversarial Attacks against Diffusion Policy:** We aim to attack robots controlled with visual-based DP, unveiling hidden threats to the safe application of diffusion-based policies. (a) By hacking the visual inputs, we can fool the diffusion process into generating wrong actions $\tau$ (in red). We propose Diffusion Policy Attacker(DP-Attacker), which can effectively attack the DP by (b) hacking the global camera inputs $I$ using small visual perturbations under both online and offline settings or (c) attaching an adversarial patch into the environment. The online settings use current visual inputs at $t$-th timestep $I^t$ to generate time-variant perturbations $\delta^t$, while the offline settings use only offline data $I^{\mathcal{D}}$ to generate time-invariant perturbations $\delta$.

remains largely unexplored, posing a potential barrier and risk to their broader application. While it is straightforward to attack an end-to-end DNN by applying gradient ascent over the loss function [31, 14], it is non-trivial to craft attacks against a DP, due to its concatenated denoising structure and high randomness. Prior research [25, 24, 56, 55, 45] has focused on attacking the diffusion process of the text-to-image (T2I) diffusion models [42]. However, there are distinct differences between attacking a T2I diffusion model and attacking a Diffusion Policy. Firstly, they concentrate on attacking the diffused value while we aim at attacking the conditional image. In addition, they try to fool the editing process over the clean images (e.g. SDEdit [34]), while we are trying to fool the robot to make wrong actions step by step, each action is generated from the pure gaussian noise. Diffusion policies are also more interactive with the environment. A successful attack not only needs to fool a single inference output but also needs to continuously fool the model to decrease model performance.

In this paper, we focus on crafting adversarial attacks against DP. Specifically, we propose Diffusion Policy Attacker (DP-Attacker), the first suite of white-box-attack algorithms that can effectively deceive the visual-based diffusion policies. We investigate two hacking scenarios as illustrated in Figure 1: (1) digital attack–hacking the scene camera, which means that we can add imperceptible digital perturbations to the visual inputs of DP, and (2) physical attack–hacking the scene by attaching small adversarial patches [4] to the environments (e.g. table). Furthermore, we consider both offline and online settings, for online settings, we can generate time-variant perturbations based on the current visual inputs, on the opposite, for the offline settings we can only add one fixed perturbation across all the frames.

We conducted extensive experiments on DP pre-trained on six robotic manipulation tasks and demonstrated that DP-Attacker can effectively craft adversarial attacks against DP. For digital attacks, DP-Attacker can generate both online and offline attacks that significantly degrade the DP system's performance. For physical attacks, DP-Attacker is capable of creating adversarial patches tailored for each task, which can be put into the physical environment to disrupt the system. Also, we reveal that the non-robust image encoder makes the DP easy to attack.

## 2 Related Works

**Diffusion-based Policy Generation** Diffusion models [49, 16, 48] exhibit superior performance in multiple domains like high-fidelity image generation, video generation and 3D generation [42, 39, 44, 47, 53, 41, 27]. Due to its strong expressiveness in modeling multi-modal distribution, diffusion models have also been successfully applied to robot learning areas such as reinforcement learning [54, 2], imitation learning [9, 58, 20, 38], and motion planning [43, 30, 18]. Among them, Diffusion policy (DP) [9, 58, 23] has gained significant attention due to its straightforward training

methodology and consistent, reliable performance. In this paper, we focus on crafting adversarial attacks against visual-based DP, a technology already integrated into various indoor robot prototypes like Mobile Aloha [12].

**Adversarial Examples for Deep Systems**  Adversarial attacks have been widely studied for deep neural networks (DNNs): given a small perturbation, the DNN will be fooled to make wrong predictions [51, 14]. For DNN-based visual recognition models, crafting adversarial samples is a relatively easy task using gradient-based budget-limited attacks [31, 57, 14, 5, 10, 3]. However, attacking diffusion models consisting of a cascade of DNNs injected with noise, poses a more complex challenge. Recent studies have demonstrated the feasibility of effectively crafting adversarial samples for latent diffusion models using meticulously designed surrogate losses [25, 59, 24, 46, 45, 56, 6]. However, these efforts have primarily focused on image editing or imitation tasks and are limited to working solely in latent space [55]. Here we hope to explore the adversarial attacks against DP under various settings.

**Adversarial Threats against Robot Learning**  Previous research has highlighted adversarial attacks as a significant threat to robot learning systems [8], where small perturbations can cause chaos in applications such as deep reinforcement learning [22, 13, 28, 37, 50, 36], imitation learning [15], robot navigation [21], robot manipulation [19, 33], and multi-agent robot swarms [1]. Despite the rising popularity of policies generated by diffusion models, to the best of our knowledge, there have been no prior efforts aimed at attacking these models in the field of robotics.

# 3 Preliminaries

## 3.1 Diffusion Models for Behaviour Cloning

Diffusion models [49, 16] are one type of generative model that can fit a distribution $q(x_0)$, using a diffusion process and a denoising process. Starting from $x_K$, a pure Gaussian noise, the denoising process can generate samples from the target distribution by $K$ iterations of denoising steps (Here we use $K, k$ to represent steps in diffusion and $T, t$ for running timesteps of the robot scenarios):

$$x_k = \alpha_k(x_{k+1} - \lambda_k \epsilon_\theta(x_k, k) + \mathcal{N}(0, \sigma_k^2 I)), \ k = 0, 2, ..., K-1 \tag{1}$$

where $\alpha_k, \lambda_k, \sigma_k$ are hyper-parameters for the noise scheduler. $\epsilon_\theta$ is a learned denoiser parameterized by $\theta$, which can be trained by optimizing the denoising loss termed $\mathcal{L} = \mathbb{E}_{x,k} \|\epsilon_\theta(x + \epsilon_k, k) - \epsilon_k\|^2$. We define the reverse process in Equation 1 as $x_k = \mathcal{R}_\theta^k(x_{k+1})$ for simplicity.

Diffusion policies [18, 9] noted $\pi_\theta$ apply the diffusion models mentioned above, resulting in $\tau^t \sim \pi_\theta(s^t)$, where $\tau^t \in \mathbb{R}^{D_a \times L_a}$ is the planned action sequences at timestep $t$ in the continuous space, $s^t$ is the current states, and $D_a, L_a$ are the action dimension and action length respectively. Accordingly, the learnable denoiser becomes $\epsilon_\theta(\tau_k, k, s)$, and the denoised diffusion process remains the same. For visual DP, the states $s^t$ are usually images captured by the scene or wrist cameras, so we use $I^t$ throughout to represent the visual inputs at timestep $t$. Finally, the policy can be formulated as

$$\tau^t \sim \pi_\theta(I^t) = \mathcal{R}_\theta^0(\mathcal{R}_\theta^1...\mathcal{R}_\theta^{K-2}(\mathcal{R}_\theta^{K-1}(x_K, I^t)...I^t), I^t). \tag{2}$$

The equation above shows that the predicted action $\tau^t$ is the output of chained denoiser models residually conditioned on the current observation $I^t$. In practice, while DP outputs a long sequence of actions $\tau$, we only execute the first few actions of it in a receding horizon manner to improve temporal action consistency [9].

## 3.2 Adversarial Attacks Against Diffusion Models

Adversarial samples [14, 31, 5] have been widely studied as a threat to the AI system: for a DNN-based image classifier $y = f_\theta(x)$, one can easily craft imperceptible perturbations $\mathcal{P}$ to fool the classifier to make wrong predictions over $\mathcal{P}(x)$. In digital attack settings [51, 14], the perturbation should be small and always invisible to humans, which can be formulated by the $\ell_\infty$-norm as $|\mathcal{P}(x) - x|_\infty < \sigma$ where $\sigma$ is a small value (e.g. $8/255$ for pixel value). Methods like FGSM [14] and PGD [31] can be easily applied to craft such kinds of adversarial attacks. For physical-world

adversarial patches [4, 11, 57, 17], $\mathcal{P}(x)$ is always crafted as attaching a small adversarial patch to the environments, and the patch should be robust to physical-world transformations such as position, camera view, and lighting condition.

Recent works [25, 45] show that it is also possible to craft such kind of adversarial examples to fool latent diffusion models [42] with an encoder $\mathcal{E}$ and a decoder $\mathcal{D}$: adding small perturbation to a clean image, the denoising process will be fooled to generate bad editing or imitation results. The following Monte-Carlo-based adversarial loss to attack a latent diffusion model:

$$\mathcal{L}_{adv}(x) = \mathbb{E}_k \|\epsilon_\theta(\mathcal{E}(x) + \epsilon_k, k) - \epsilon_k\|_2^2 \tag{3}$$

the mechanism behind attacking latent diffusion models [56] turns out to be the vulnerability of the autoencoder and works only for the diffusion model in the latent space [55]. Also, the settings above differs from our settings of attacking a DP which targets on attacking the conditional image without the ground-truth clean action to get the diffused input of $\epsilon_\theta$ in Equation 3. In the following section, we show that we can still effectively craft different kinds of adversarial samples based on Equation 3 with some modification.

# 4 Methods

## 4.1 Problem Settings

**Threat Model** In this paper, we assume that we have white-box access to some diffusion policy network. That is, we have access to its parameters and also the data used to train it. Given this trained network, we wish to find adversarial perturbations that, when added to the observation $I$, will cause the trained diffusion policy to generate unwanted actions (either random or targeted) that impede task completion (lower the task score or success rate). We consider two types of perturbations detailed in Sec. 4.2 and Sec. 4.3.

The most straightforward way to measure the quality of the attack is to use the difference between generated actions from the original actions in an end-to-end manner:

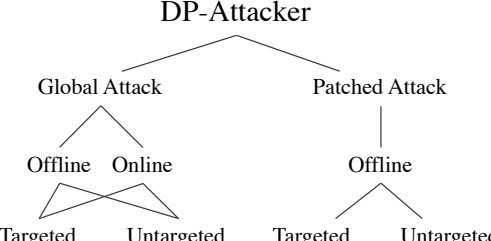

Figure 2: **Design Space of** `DP-Attacker`: the tree above shows the design space of `DP-Attacker`, which can be adapted to various kinds of attack scenarios, including global attacks (hacking and cameras) vs patched attacks (hacking the physical environment); offline vs online; targeted vs untargeted.

$$\mathcal{L}_{\text{end2end}}^{\text{untar}}(I, t) = -\|\pi_\theta(\mathcal{P}(I)) - \tau^{t,*}\|^2 \tag{3}$$

where $\tau^{t,*}$ is a known good solution sampled by $\pi_\theta$ given the observation image $I$, and $\mathcal{P}(\cdot)$ is some perturbation on the observation image. It could be generated either from the trained policy for online attacks or from the training dataset for offline attacks. One can minimize the negative L-2 distance between a generated action and a good action for untargeted attacks. For targeted attacks, the action loss becomes

$$\mathcal{L}_{\text{end2end}}^{\text{tar}}(I, t) = \|\pi_\theta(\mathcal{P}(I)) - \tau_{\text{target}}^t\|^2 \tag{4}$$

where $\tau_{\text{target}}^t$ is some target bad action we wish the policy to execute (e.g. always move to left). We can use PGD [31] to optimize for the best perturbation that minimizes this loss. However, due to the inherent long-denoising chain of the diffusion policy $\pi_\theta$, the calculation of this gradient could be quite costly [45].

In practice, running the end-to-end attacks above is not effective especially when the model is large and when we need to hack the camera at a high frequency. Instead, borrowing ideas from recent works [25, 24, 56] on adv-samples for diffusion models, we propose to use the following optimization objectives:

$$\mathcal{L}_{\text{adv}}^{\text{untar}}(I, t) = -\mathbb{E}_k \|\epsilon_\theta(\tau^{t,*} + \epsilon_k, k, \mathcal{P}(I)) - \epsilon_k\|^2 \tag{5}$$

where $k$ is the timestep of the diffusion process and $t$ is the timestep of the action runner. We add noise to the good solution $\tau^{t,*}$ and then calculate the L-2 distance between the predicted noise of the

denoise network and the added noise. Minimizing this loss leads to inaccurate noise prediction of the denoising network and, in turn, leads to bad generated action of the diffusion policy. For targeted attacks, the noise prediction loss is:

$$\mathcal{L}_{\text{adv}}^{\text{tar}}(I, t) = \mathbb{E}_k \|\epsilon_\theta(\tau_{\text{target}}^t + \epsilon_k, k, \mathcal{P}(I)) - \epsilon_k\|^2 \tag{6}$$

Minimizing this loss would allow the denoising net to favor the generation of the target action. The gradient of the noise prediction loss is easier to calculate compared to the action loss because of the short one-step chain. This makes it more favorable for conducting attacks.

## 4.2 Global Attacks

A global attack injects adversarial perturbation $\delta$ into the observation image $I$ by adding it on top of the observation image, *i.e.* $\mathcal{P}(I) = I + \delta$. The adversarial noise $\delta$ is of the same shape as the original image. To make the attack imperceptible, the adversarial noise's absolute value is limited by some $\sigma$. To find such an adversarial noise, we use PGD [31], an optimization-based method to search for an adversarial noise. The adversarial noise can be constructed online during inference or offline using the training dataset. The algorithm for conducting an online global attack is shown in Algorithm 1. The algorithm optimizes for loss in Equation 5 or Equation 6. The algorithm can be modified easily to construct an offline attack. Given the training dataset $D_T = \{(\tau^t, I^t) | t \in T\}$ we can optimize for the loss $\mathcal{L}_{\text{adv}}^{\text{untar}}(I, t) = -\mathbb{E}_{k,(\tau^t,I^t)} \|\epsilon_\theta(\tau^t + \epsilon_k, k, \mathcal{P}(I)) - \epsilon_k\|^2$ or $\mathcal{L}_{\text{adv}}^{\text{tar}}(I, t) = \mathbb{E}_{k,(\tau^t,I^t)} \|\epsilon_\theta(\tau_{\text{target}}^t + \epsilon_k, k, \mathcal{P}(I)) - \epsilon_k\|^2$. This algorithm is provided in the appendix.

---

**Algorithm 1** Global Adversarial Attack (Online)

---

**Input:** given observation image $I$, diffusion policy $\pi_\theta$, noise prediction net $\epsilon_\theta$, attack budget $\sigma$, step size $\alpha$, number of steps $N$
**Output:** adversarial noise $\delta$
  $\delta \leftarrow 0$                                                     ▷ initialize adversarial noise
  **for** $i = 1$ to $N$ **do**
    $\epsilon_k, k \sim \mathcal{N}(0, I), \text{randint}(1, K)$                 ▷ sample forward noise and timestep
    **if** targeted attack **then**
      $\tau^t \leftarrow \tau_{\text{target}}^t + \epsilon_k$                   ▷ forward sample, $\tau_{\text{target}}^t$ should be given
      $s \leftarrow 1$
    **else if** untargeted attack **then**
      $\tau^t \sim \pi_\theta(I)$             ▷ use diffusion policy to generate a good solution
      $\tau^t \leftarrow \tau^t + \epsilon_k$                         ▷ forward sample
      $s \leftarrow -1$
    **end if**
    $\epsilon_p \leftarrow \epsilon_\theta(\tau^t, k, \text{clip}(I + \delta, 0, 1))$
    $\mathcal{L} \leftarrow s \cdot \|\epsilon_k - \epsilon_p\|^2$
    $\delta \leftarrow \text{clip}(\delta - \alpha \cdot \text{sign}(\nabla_{I_{\text{adv}}} \mathcal{L}), -\sigma, \sigma)$         ▷ Projected-Gradient Descent
  **end for**
  **return** $\delta$

---

## 4.3 Patched Attacks

A patched attack directly puts a specifically designed image patch $x \in \mathbb{R}^{c \times h \times w}$ into the environment. The camera later captures it and causes undesirable motion from the diffusion policy. The patch should be active under different scales, orientations, and observation views. During training, we apply some random affine transform (shift, rotation, scale, and shear) $\mathcal{T} \in \mathbb{T}$. The affine transform uses the center of the image as the origin of the coordinate system. The resulting patch replaces the original observation image using the replacement operator: replace($I, x$) again using the image's center as the origin of the coordinate system. To search for such a patch, we use the training dataset and optimize for the best patch using PGD. The algorithm is illustrated in Algorithm 2.

---

**Algorithm 2** Patch Adversarial Attack

---

**Input:** training dataset $D_T = \{(\tau^t, I^t)|t \in T\}$, diffusion policy $\pi_\theta$, noise prediction net $\epsilon_\theta$, set of affine transforms $\mathbb{T}$, step size $\alpha$
**Output:** adversarial patch $x$

  $x \sim \mathcal{N}(0, I)$
  $x \leftarrow \text{clip}(x + 0.5, 0, 1)$                                            $\triangleright$ initialize a patch
  **repeat**
    $(\tau^t, I^t), \mathcal{T}, \epsilon_k, k \sim D_T, \mathbb{T}, \mathcal{N}(0, I), \text{randint}(1, K)$
                                     $\triangleright$ sample from dataset, transform, forward noise, and time step
    **if** targeted attack **then**
      $\tau^t \leftarrow \tau_{\text{target}}^t + \epsilon_k$                            $\triangleright$ forward sample, $\tau_{\text{target}}^t$ should be given
      $s \leftarrow 1$
    **else if** untargeted attack **then**
      $\tau^t \leftarrow \tau^t + \epsilon_k$                                $\triangleright$ forward sample, use dataset entry
      $s \leftarrow -1$
    **end if**
    $\epsilon_p \leftarrow \epsilon_\theta(\tau^t, k, \text{replace}(I^t, \mathcal{T}(x)))$
    $\mathcal{L} \leftarrow s \cdot ||\epsilon_k - \epsilon_p||^2$
    $x \leftarrow \text{clip}(x - \alpha \cdot \text{sign}(\nabla_x \mathcal{L}), 0, 1)$             $\triangleright$ Projected-Gradient Descent
  **until** satisfied
  **return** $x$

---

## 5 Experiments

We test the effectiveness of `DP-Attacker` with various strengths and configurations on different diffusion policies. Our target models are vision-based diffusion policy models introduced by Chi et al. [9]. We aim to manipulate the visual input so that the generated trajectory will not lead to task completion. We quantitatively evaluate the effectiveness of our attack methods by recording the result task completion scores/successful rate. We also provide scores without attacks for reference and random noise attacks (adding some Gaussian noise to the observation images) as a baseline attack method. We foucus on the models released by Chi et al. [9]. However, our attack algorithm applies to other variants of diffusion policies as well.

**Environment Setup** Our benchmark contains 6 tasks: PushT, Can, Lift, Square, Transport, and Toolhang. These tasks are illustrated in Figure 7 in the Appendix. Robosuite provides all the simulation of these tasks except for PushT [52, 32, 60]. For evaluation, we attack the released checkpoints of diffusion policies trained by Chi et al. [9]. For tasks Can, Lift, Square, and Transport, each has two demonstration datasets: Multi-Human (MH) and Proficient Human (PH). The other two tasks (PushT and Toolhang) has only one PH dataset, respectively. This gives us a total of 10 datasets. In [9], each dataset is used to train two diffusion policies with different diffusion backbone architectures: CNN-based and Transformer-based. We take the best performing checkpoints for these 20 different scenarios released by Chi et. al [9] as our attack targets. For each attack method, we run 50 rollouts and collect the average score or calculate the success rate of the tasks. The rollout length uses the same length as the demonstration dataset [9, 32]. Besides our attack methods, we also run the rollout using clean images for reference and with random noise added as a baseline attack method. The evaluation is done using a single machine with an RTX 3090 GPU and AMD Ryzen 9 5950X to calculate rollouts and run our attack algorithms.

### 5.1 Global Attack

We first present the results of global attacks. We evaluate both our online attack algorithm (creating adversarial noise on the fly per inference) and offline algorithm (pre-generating a fixed noise that is used for every inference).

**Online Attack** For online attacks, we use attack parameters $\sigma = 0.03, \alpha = 0.001875, N = 50$. For targeted attacks, we use a normalized target action vector of all ones. We report the performance of the transformer-based models before and after the attack in Table 1. The results of global attacks

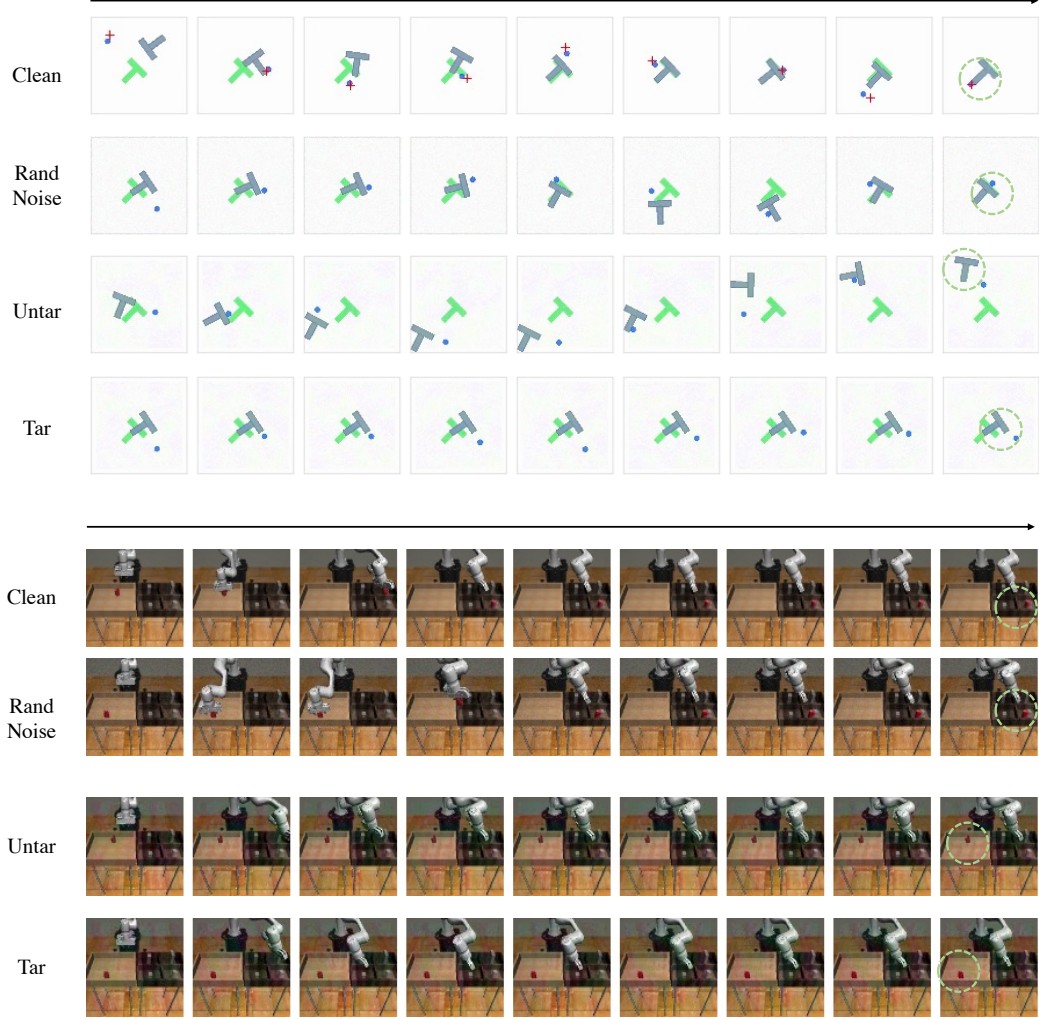

Figure 3: **Global Attack (Online)**: We visualize the global attacks in Algorithm 1 within both the PushT and Can environments. Specifically, we present action rollouts for four types of observations: clean observations, observations perturbed with random Gaussian noise, and our optimized perturbations (both untargeted and targeted). While the DPs show robustness to random perturbations, they are vulnerable to adversarial samples generated using `DP-Attacker`.

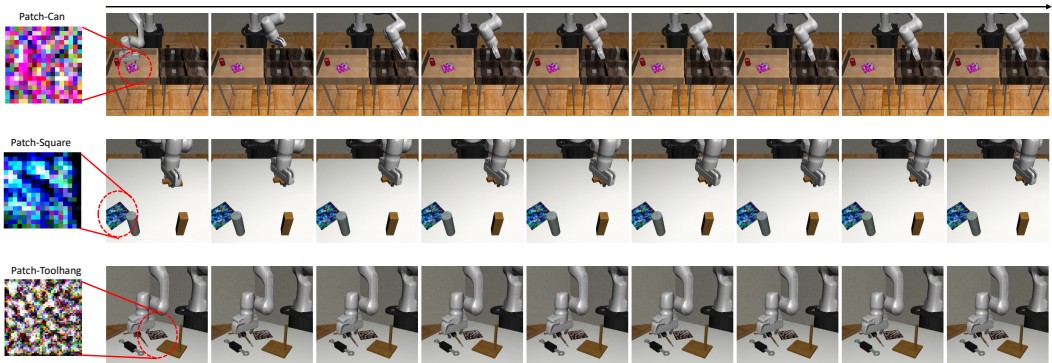

Figure 4: **Physical Adversarial Patches:** we show the patches optimized by Algorithm 2, attaching it to the physical scene will effectively lower the success rate of the target diffusion policy.

| Tasks | PushT | Can | | Lift | | Square | | Transport | | Toolhang |
| Demonstration Type | PH | PH | MH | PH | MH | PH | MH | PH | MH | PH |
|---|---|---|---|---|---|---|---|---|---|---|
| Clean | 0.75 | 0.92 | 0.92 | 1 | 1 | 0.92 | 0.72 | 0.86 | 0.46 | 0.86 |
| Random Noise | 0.66 | 0.88 | 0.98 | 1 | 1 | 0.82 | 0.74 | 0.84 | 0.48 | 0.82 |
| Targeted-Offline | 0.46 | 0.08 | 0.08 | 0.94 | 0.7 | 0 | 0 | 0 | 0.02 | 0 |
| Untargeted-Offline | 0.39 | 0.1 | 0.46 | 0.8 | 0.62 | 0.04 | 0 | 0 | 0 | 0 |
| Targeted-Online | 0.10 | 0 | 0 | 0.02 | 0 | 0 | 0 | 0 | 0 | 0 |
| Untargeted-Online | 0.19 | 0.02 | 0.02 | 0.62 | 0.62 | 0 | 0 | 0 | 0 | 0 |

Table 1: **Quantitative Results on Global Attacks:** The table includes the attack result for all transformer based diffusion policy networks. Our DP-Attack can significantly lower the performance of the diffusion models.

| | Can | | Lift | | Square | | Toolhang | |
| Backbone Arch | CNN | Transformer | CNN | Transformer | CNN | Transformer | CNN | Transformer |
|---|---|---|---|---|---|---|---|---|
| Clean | 0.98 | 0.92 | 1 | 1 | 0.94 | 0.92 | 0.8 | 0.86 |
| Random Noise Patch | 0.9 | 0.94 | 1 | 0.9 | 0.8 | 0.54 | 0.56 | 0.12 |
| Untargeted-Offline | 0.16 | 0.44 | 1 | 0.82 | 0.72 | 0.34 | 0.48 | 0.02 |

Table 2: **Quantitative Results on Patched Attacks**

on all models are given in the appendix. Example rollouts and images used in the rollouts are shown in Figure 3.

**Offline Attack** For offline global attacks, we train on the training dataset with batch size 64, $\alpha = 0.0001, \sigma = 0.03$ for 10 epochs. The resulting trained adversarial noise is added to the input image for every inference. The results are shown in Table 1. Examples of rollouts and images used in the attack can be found on our website.

We find that diffusion policy is not robust to noises introduced by our DP-Attacker. The performance of diffusion policies is significantly reduced after running global attacks. A disturbance of less than 3% is able to decrease the performance from 100% to 0%. The success of offline global attacks also shows attacks can be cheaply constructed and pose a significant threat to the safety of using diffusion policy in the real world.

## 5.2 Patched Attack

**Training vs. Evaluating** Since patched attacks directly put an attack image into the environment, we only consider offline attacks that pre-generate some patch that is used throughout the rollout. We train the patch using Algorithm 2, where the patch is applied to the training image using some randomized affine transform. This allows the gradient to pass through for successful training. Since we have used random affine transforms during training, the patch should be transferable when used in the simulation environment. For evaluation, we create a thin box object with the trained image patch as its texture and put it randomly onto the table.

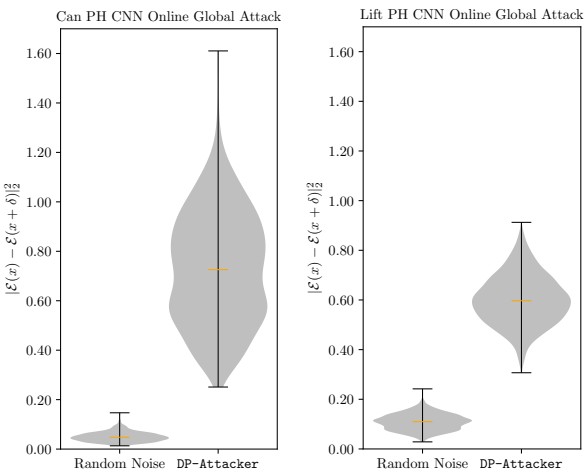

Figure 5: **Difference in Encoded Feature Vector**: we calculate the distance between the clean feature vector and the attacked feature vector. DP-Attacker perturb the feature vector significantly compared to naive random noise attack.

| Parameters ($\sigma = 0.03$) | $N = 10$ | $N = 20$ | $N = 50$ |
|:---:|:---:|:---:|:---:|
| Success Rate | 0.94 | 0.8 | 0.66 |
| Parameters ($N = 50$) | $\sigma = 0.01$ | $\sigma = 0.03$ | $\sigma = 0.05$ |
| Success Rate | 1 | 0.68 | 0.32 |

Table 3: **Different Parameters for** `DP-Attack`**:** We did an ablation study on parameters $\sigma$ and $N$, and we can see that smaller steps and budgets are not enough to fool a DP. Larger budgets will dramatically decrease the Sucess Rate (SR).

**Results** We construct a patch of size that covers around 5% of the observation image using Algorithm 2. The details of the training can be found in the appendix. We evaluate the effectiveness of our patch attack algorithm on a total of 8 checkpoints, covering the PH dataset across four tabletop manipulation tasks (Can, Lift, Square, and Toolhang) using both CNN and Transformer diffusion backbones. The result success rate (SR) is shown in Table 2. Example rollouts are shown in Table 4.

Simpler tasks such as Can and Lift are quite robust to random noise patch. Our `DP-Attacker` produces adversarial patches that perform better than random noise in terms of degrading the diffusion policy performance.

### 5.3 Quantitative Results on Targeted Attacks

We qualitatively evaluate the effectiveness of our targeted attacks. We use our `DP-Attacker` to run global-online-targeted attacks with varying strength on two model checkpoints: PushT (CNN) and CAN (PH CNN). The target in PushT task is a 2D coordinate around $(323.875, 328.75)$ (note the side length of the PushT environment is 1024), and the target in the CAN task is the target end-effector position around $(0.1686, 0.1049, 1.0848)$ (in meters). In 6, we report how close the actions generated by diffusion policy are to our attack targets during the rollout.

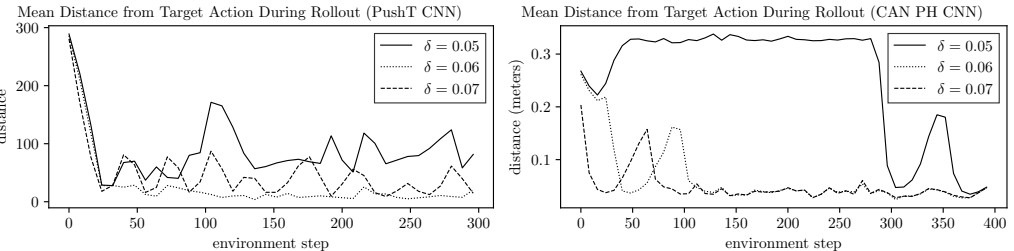

Figure 6: We used DP-Attacker with different attack strengths to run rollouts. We report the average distance between the predicted action sequence and the target action sequence (a sequence of a duplicate target coordinate). The metric is calculated at each inference during the rollout.

Our `DP-Attacker` is able to manipulate the generated action to be within 20 units and 5 cm of the attack targets, respectively, for the PushT task and CAN task, with an attack strength of $\delta = 0.06$. Example rollouts of these two attack scenarios can be found in the Sec. D of the appendix.

## 6 Ablation Study

**Attack Parameters** To investigate the effectiveness of our attack method, we evaluate how the attack parameter plays a role in `DP-Attacker`. First, we investigate the effect of the number of PGD steps $N$. We keep the $\sigma = 0.03$, and $\alpha = \frac{2\sigma}{N}$. Second, we investigate the effect of the noise scale $\sigma$. We keep $N = 50$, and $\alpha = \frac{2\sigma}{N}$. We evaluate all six attacks on the transformer backbone DP trained on the Lift PH dataset. The result is summarized in table 3.

**End to End Loss vs. Noise Prediction Loss** We perform a comparison with the end-to-end action loss 3. We evaluate both methods with the same attack parameters ($\sigma = 0.03, \alpha = 0.001875, N = 50$) on the best-performing transformer backbone trained on the PH dataset of the Lift task. Again, we evaluate 50 randomly initialized environments. The selection of end to end loss with DDPM [16]

| Method | Attack Time | Model Success Rate |
|---|---|---|
| Clean | - | 1 |
| Random Noise | - | 1 |
| End to End DDPM (Untargeted) | $\sim$70s | 1 |
| End to End DDPM (Targeted) | $\sim$67s | 0.52 |
| End to End DDIM-8 (Untargeted) | $\sim$6.5s | 0.9 |
| End to End DDIM-8 (Targeted) | $\sim$5.8s | 0.24 |
| DP-Attacker (Untargeted) | $\sim$1.8s | 0.62 |
| DP-Attacker (Targeted) | $\sim$1.3s | 0.02 |

Table 4: **Compared with End to End Attacks** `DP-Attacker` runs significantly faster than the end-to-end attacks even if it is accelerated with DDIM. Our `DP-Attacker` also provides better attack results.

scheduler makes it infeasible for online attacks. In addition, we provide results where we replace the loss-calculating noise scheduler with a DDIM-8 step scheduler [48]. This provides speedup for calculating the end to end loss. The result SR after the attack and the average time used to perform the online attacks are shown in 4. The naive end-to-end loss is significantly lower than our attack algorithms and does not provide better results. We suspect that since diffusion models introduce randomness during the sampling of a trajectory, it is better to attack the noise prediction loss rather than the end to end action loss.

**What Is Being Attacked Is the Encoder**  We try to investigate further what exactly is attacked in our `DP-Attacker`. Other literature relating to text-to-image diffusion models shows that the encoder is the one being attacked [45, 56]. We suspect the same is happening for diffusion policy. To investigate this, we calculate the L2 distance between the encoded feature vector of clean and attacked images random noise attack, unsuccessful attack parameters, and successful parameters, respectively. The details of the calculation is in the appendix. We do this for 1000 images in the training dataset and plot the distribution of the distances using a violin plot in Figure 5. The significant difference shows that our attack method has drastically changed the representation of the conditional visual feature. This later affects the downstream conditional noise prediction net, causing it to make inaccurate noise predictions. We put details about it in the Appendix.

## 7 Conclusion and limitations

In this paper, we propose `DP-Attacker`, a suite of algorithms designed to effectively attack diffusion-based policy generation, an emerging approach in behavior cloning. We demonstrate that `DP-Attacker` can craft adversarial examples across various scenarios, posing a significant threat to systems reliant on DP. Our findings highlight that despite the inherent randomness and cascaded deep structure of diffusion-based policy generation, it remains vulnerable to adversarial attacks. We emphasize the need for future research to focus on enhancing the robustness of DP to ensure its reliability in real-world applications. There are also some limitations for this paper: our experiments were conducted exclusively within a simulation environment, and we did not extend our testing to real-world scenarios. Additionally, we did not develop or implement any defensive strategies for the proposed tasks, which remains an area for future research and exploration.

## Acknowledgments and Disclosure of Funding

We thank the anonymous reviewers for their valuable feedback on this paper. This work is supported in part by grants NSF CAREER ECCS-1942523 and NSF FRR-2409016.

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

# Appendix

We put more video results, including rollouts of the DP-based robots under various attacks crafted by `DP-Attacker` , in the following anonymous link:

## A    Broader Impact

Diffusion-based policies (DPs) have emerged as promising candidates for integrating real robots into our daily lives. Even with just a few collected demonstrations, DPs exhibit strong performance across various tasks [12, 58]. However, despite their utilization of diffusion models, which distinguish them from other policy generators, our research highlights their vulnerability to adversarial attacks. We demonstrate practical attacks on DP-based systems, such as hacking cameras to introduce fixed perturbations across all frames (global offline attack) and incorporating patterns into the scene (physical patched attack). It is critical to consider these threats, and we urge future research to prioritize the development of more robust DPs before their widespread application in the real world.

## B    Algorithms

We also provide the algorithm for training the offline global attacks.

---

**Algorithm 3** Global Adversarial Attack (Offline)

---

**Input:**  training dataset $D_T = \{(\tau^t, I^t)|t \in T\}$, diffusion policy $\pi_\theta$, noise prediction net $\epsilon_\theta$, step size $\alpha$, attack budget $\sigma$
**Output:**  adversarial noise $\delta$
   $\delta \leftarrow 0$                                                            ▷ initialize adversarial noise
   **repeat**
      $(\tau^t, I^t), \epsilon_k, k \sim D_T, \mathcal{N}(0, I), \text{randint}(1, K)$ ▷ sample from dataset, forward noise and timestep

      **if** targeted attack **then**
         $\tau^t \leftarrow \tau^t_{\text{target}} + \epsilon_k$                         ▷ forward sample, $\tau^t_{\text{target}}$ should be given
         $s \leftarrow 1$
      **else if** untargeted attack **then**
         $\tau^t \leftarrow \tau^t + \epsilon_k$                                    ▷ forward sample
         $s \leftarrow -1$
      **end if**
      $\epsilon_p \leftarrow \epsilon_\theta(\tau^t, k, \text{clip}(I^t + \delta, 0, 1))$
      $\mathcal{L} \leftarrow s \cdot ||\epsilon_k - \epsilon_p||^2$
      $\delta \leftarrow \text{clip}(\delta - \alpha \cdot \text{sign}(\nabla_{I_{\text{adv}}} \mathcal{L}), -\sigma, \sigma)$         ▷ Projected-Gradient Descent
   **until** satisfied
   **return**  $\delta$

---

## C    Experimental Details

### C.1    Task Descriptions

We investigate a total of six different tasks: PushT, Can, Lift, Square, Transport, and Tool hang. The tasks are illustrated in 7. Here are the descriptions for each task:

- PushT: The simulation happens in 2D. The agent controls a rod (blue circle) to push the grey T block into the targeted green area. The score calculated is the maximum percent of coverage of the green area by the grey T block during a rollout.

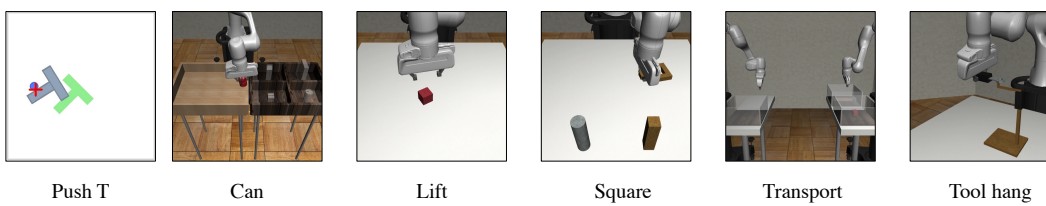

| Push T | Can | Lift | Square | Transport | Tool hang |

Figure 7: **Simulation renders of the six tasks**

- Can: The simulation environment is provided by robosuite [60]. The agent controls the 6-DoF end-effector position and gripper close or open. The goal is to move the randomly positioned can in the left bin into the corresponding bin (lower right) in the right bin.
- Lift: The simulation environment is provided by robosuite. The agent controls the 6-DoF end-effector position and gripper close or open. The goal is to lift up the randomly positioned red block.
- Square: The simulation environment is provided by robosuite. The agent controls the 6-DoF end-effector position and gripper close or open. The goal is to put the randomly positioned square nut around the square peg.
- Transport: The simulation environment is provided by robosuite. The agent controls 2 6-DoF end-effector positions and grippers close or open. The goal is to transport the hammer inside the box one one side to the box on the other side.
- Tool hang: The simulation environment is provided by robosuite. The agent controls the 6-DoF end-effector position and gripper close or open. The goal is to construct the tool tower by first inserting an L-shaped bar into the base and later hanging the second tool on the tip of the bar.

### C.2   Training Details

Our goal is to construct noises for the observation images. The image encoder uses multiple views when constructing the conditional image feature vector. Below are details of how we construct the adversarial noises.

#### C.2.1   Global Online Attacks

For global online attacks, we construct noises for all observation views, i.e. suppose the encoder takes two camera views (one side view and one eye-in-hand), and the conditional state length is two, we will construct a total of four noises for adding onto the input images, respectively, before passing it into the policy for action generation.

For the random noise attack, the noise selected is sampled from a standard Gaussian and scaled by $\sigma = 0.03$ and clipped in the range $[-\sigma, \sigma]$. For untargeted online attacks, we use PGD parameters $N = 50, \sigma = 0.03, \alpha = 0.001875$. For targeted online attacks, the targeted selected is an action matrix (actim dim $\times$ action horizon) of all 1's (in normalized action space). The PGD parameters for targeted online attacks are the same as the untargeted online attacks.

#### C.2.2   Global Offline Attacks

Similar to global online attacks, we also construct noise for all observation views. However, since this is an offline attack, we pre-generate (train) just one set of adversarial noises for each input, and it is used throughout the rollout for the same task. The training parameters for untargeted and targeted attacks are the same: number of epochs = 10, $\alpha = 0.0001$, and batch size = 64. For targeted attacks, we again use a normalized action of all 1's.

#### C.2.3   Patched Attacks

For patch attack training, we only apply the patch on the most important camera view (side view). However, this is to maintain the consistency for the patch gradient pass. However, the patch is put into

| Parameter | Range |
|---|---|
| x shift | $[-0.4, 0.4]$ percent of image, in centered coordinate. |
| y shift | $[-0.4, 0.4]$ percent of image, in centered coordinate. |
| rotation | $[-45°, 45°]$ |
| scale | $[1, 1]$ |
| shear x | $[-50°, 50°]$ |
| shear y | $[-50°, 50°]$ |

Table 5: **Set of Affine Transforms $\mathbb{T}$ for Patched Attack**

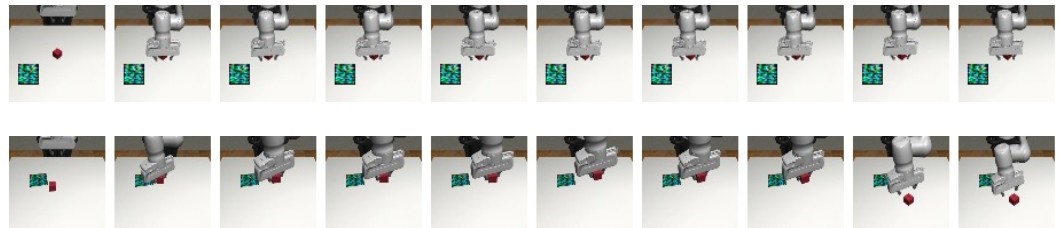

Figure 8: **Comparison of Physical Patch Training and Evaluation**

the simulation environment for evaluation and can be observed from multiple perspectives. For tasks Can, Lift, and Square, the observation image size is $84 \times 84$, and we choose the training patch size of $17 \times 17$ that covers around 4% of the observation. For task Toolhang where the observation image size is $84 \times 84$, we choose the training patch size of $50 \times 50$ that covers around 4.3% of the image. The set of transforms $\mathbb{T}$ is summarized in the table 5. The training parameters are epochs = 10, batch size = 64, $\alpha = 0.0001$. For evaluation, we make patch objects of size $0.06\text{m} \times 0.06\text{m}$ ($2.36\text{in} \times 2.36\text{in}$) and put it onto the table. The rotation angle is from $[-45°, 45°]$. For tasks Can, Lift, and Square, the position of the patch can be anywhere on the table. For Toolhang, the position of the patch is constrained to be on the top left of the table so it can be captured by the camera. The size is about the same and we provide the comparison in the figure 8.

# D More Results

**Full Table for Global Attacks**   We provide the full table for the results of global attacks in 6 as an extension for 1. CNN-based models are harder to attack. Nevertheless, the result score still decreased significantly.

**Targeted Attacks**   With more attack budget, we can manipulate the robot's action quite well. For this experiment, we increase the online global attack budget $\sigma$ to 0.05. With this increased budget, we could manipulate the generated action of DP. This shows the effectiveness of `DP-Attacker` proposed targeted noise prediction loss. See our website for details.

**What Is Being Attacked Is the Encoder**   To investigate whether it is the encoder that is being attacked in `DP-Attacker` we perform the following comparison. For a given image, we find the encoded feature vector of the clean image $\mathcal{E}(x)$, clean image + random noise $\mathcal{E}(x + \delta_{\text{rand}})$, and clean image + adversarial noise $\mathcal{E}(x + \delta_{\text{adv}})$ by our `DP-Attacker`. Next we calculate the L2 distance between the encoded clean image vs the encoded random noise attacked image $|\mathcal{E}(x) - \mathcal{E}(x + \delta_{\text{rand}})|_2^2$, and the L2 distance between the encoded clean image vs the encoded `DP-Attacker` attacked images $|\mathcal{E}(x) - \mathcal{E}(x + \delta_{\text{adv}})|_2^2$. We collect the these two distances for 1000 images in the training dataset and plot the distribution of the two sets using violin plots 5. The attack we use are random noise attack with $\sigma = 0.03$ and online targeted global attacks with $\sigma = 0.03$, $N = 50$, $\alpha = 0.001875$, ntarget = **1**. We do this for two dataset, one is the CAN ph with CNN backbone where our `DP-Attacker` successfully performs the attack, and the other is the Lift ph with CNN backbone where our `DP-Attacker` fails to construct successful attacks (see table 6). The difference in distribution shows that successful attacks is correlated with the successful disturbance of the encoder.

| Tasks | PushT | Can | | Lift | | Square | | Transport | | Toolhang |
| Demonstration Type | PH | PH | MH | PH | MH | PH | MH | PH | MH | PH |
|---|---|---|---|---|---|---|---|---|---|---|
| Clean | 0.75/0.83 | 0.92/0.98 | 0.92/0.98 | 1/1 | 1/1 | 0.92/0.94 | 0.72/0.84 | 0.86/0.88 | 0.46/0.82 | 0.86/0.8 |
| Random Noise | 0.66/0.87 | 0.88/1 | 0.98/0.98 | 1/1 | 1/1 | 0.82/0.94 | 0.74/0.76 | 0.84/0.84 | 0.48/0.68 | 0.82/0.72 |
| Targeted-Offline | 0.46/0.68 | 0.08/0.2 | 0.08/0.16 | 0.94/1 | 0.7/1 | 0/0.9 | 0/0.66 | 0/0.66 | 0.02/0.64 | 0/0 |
| Untargeted-Offline | 0.39/0.73 | 0.1/0 | 0.46/0.34 | 0.8/1 | 0.62/0.98 | 0.04/0.62 | 0/0.68 | 0/0 | 0/0 | 0/0 |
| Targeted-Online | 0.10/0.45 | 0/0 | 0/0 | 0.02/1 | 0/1 | 0/0.54 | 0/0.08 | 0/0 | 0/0 | 0/0 |
| Untargeted-Online | 0.19/0.48 | 0.02/0.02 | 0.02/0.02 | 0.62/1 | 0.62/1 | 0/0.38 | 0/0.08 | 0/0.04 | 0/0.04 | 0/0 |

Table 6: **Quantitative Results on Global Attacks:** The table includes the attack results for both CNN and transformer-based diffusion policy networks. The format is transformer/CNN. Our `DP-Attack` can significantly lower the performance of the diffusion models.

| Method | Attack Time | Model Score |
|---|---|---|
| Clean | - | 0.75 |
| Random Noise | - | 0.66 |
| End to End DDPM | ∼44.6s | 0.11 |
| End to End DDIM-8 | ∼4.3s | 0.11 |
| `DP-Attacker` Targeted-Online | ∼1.5s | 0.11 |

Table 7: **Compared with End to End Attacks** `DP-Attacker` runs significantly faster than the end-to-end attacks even if it is accelerated with DDIM. Our `DP-Attacker` also provides better attack results

**Speed and Effectiveness Comparison With End to End Loss** We perform another set of comparison with the end to end loss to show the both the speed benefit and effectiveness of our `DP-Attacker`. We conduct online targeted attacks on the Transformer-based DP for the PushT task. The PGD parameters for end to end attacks at $N = 50, \sigma = 0.03, \alpha = 0.001875$. The result average model score in 50 simulations and attack time is shown in 7. The evaluation is done on a machine with RTX4080 mobile GPU , and Intel i9-13900HX CPU.

**Rollouts of Quantitative Targeted Attacks** The following are the rollouts of videos corresponding to Sec. 5.3. Please open this PDF with Adobe Acrobat Reader to view the animated frames.

Figure 9: $\delta = 0.05$ (PushT)    Figure 10: $\delta = 0.06$ (PushT)    Figure 11: $\delta = 0.07$ (CAN PH)

Figure 12: $\delta = 0.05$ (CAN PH) Figure 13: $\delta = 0.06$ (CAN PH) Figure 14: $\delta = 0.07$ (CAN PH)

**Transferability of Offline Attacks Across Different Backbones**    We also evaluate the transferability of DP-Attacker generated offline attacks across different diffusion backbones. We first test the transferability of offline global attacks. We use DP-attacker to generate untargeted global offline attacks ($\delta = 0.03$) on two checkpoints (CAN-MH-CNN and CAN-MH-TF). Then, we evaluate these models using the two generated adversarial perturbations, and the success rate is listed in 8. We also test the transferability of patched attacks. We use DP-Attacker to generate adversarial patches on two checkpoints (CAN-PH-CNN and CAN-PH-TF). Then, we evaluate these models using the two generated adversarial perturbations, and the success rate is listed in Table 9.

| Attacked Model / Runner Model | CNN | TF | Original SR |
|---|---|---|---|
| CNN | 0.34 | 0.78 | 0.98 |
| TF | 0.32 | 0.46 | 0.92 |

Table 8: Model success rate of the CAN task trained on the MH dataset. We generated offline global attacks for each backbone first, and tested them on both models. The attack transfer case is marked in blue. We also report the model's original SR (without perturbation).

| Attacked Model / Runner Model | CNN | TF | Original SR |
|---|---|---|---|
| CNN | 0.16 | 0.54 | 0.98 |
| TF | 0.42 | 0.44 | 0.92 |

Table 9: Model success rate of the CAN task trained on the PH dataset. We generated physical patched attacks for each backbone first, and tested them on both models. The attack transfer case is marked in blue. We also report the model's original SR (without perturbation).

Note that from 2, the random noise patch does not affect the performance of DP on the CAN task very much. However, our DP-Attacker generated patches are able to decrease model performance in transfer settings (marked in blue) in Tables 8 and 9. This shows the effectiveness of our DP-Attacker and its potential ability in black-box attacks.

