# OpenReview forum: "Diffusion Policy Attacker: Crafting Adversarial Attacks for Diffusion-based Policies"
_NeurIPS.cc/2024/Conference — NeurIPS 2024 poster_

### Official Review · Reviewer_7kiH · 2024-07-11

**Soundness:** 3
**Presentation:** 3
**Contribution:** 3
**Rating:** 7
**Confidence:** 4

**Summary:**

This paper introduces DP-Attacker, a suite of algorithms designed to generate adversarial attacks against diffusion-based policies (DPs). The paper explores two attack scenarios: (1) hacking the scene camera by adding imperceptible digital perturbations to the visual inputs, and (2) hacking the scene by attaching small adversarial patches to the environment. The authors demonstrate the effectiveness of DP-Attacker through extensive experiments on pre-trained diffusion policies across various robotic manipulation tasks, showing significant degradation in performance for both online and offline attacks.

**Strengths:**

- Novelty: The paper presents the first suite of white-box attack algorithms specifically designed for visual-based diffusion policies. The proposed approach, based on noise prediction loss, effectively circumvents the challenges posed by the chained denoising structure and high randomness of diffusion models.

- Significance: By highlighting the vulnerability of diffusion policies to adversarial attacks, the paper raises important safety concerns for real-world applications. This research serves as a crucial step towards developing more robust DP systems and ensuring their reliability in practical scenarios.

**Weaknesses:**

- Lack of Defense Strategies: While the paper demonstrates the effectiveness of DP-Attacker, it does not explore or propose any defensive strategies to mitigate the identified vulnerabilities. This omission hinders the development of robust DP systems and leaves open the question of how to protect against such attacks.
- Limited Attack Scenarios: The paper focuses on two specific attack scenarios: hacking the camera and attaching adversarial patches. Exploring a broader range of attack scenarios, such as manipulating the robot's sensors or exploiting other weaknesses in the DP system, would provide a more comprehensive understanding of the system's vulnerabilities.
- Computational Complexity: The paper mentions the computational complexity of gradient calculations for the end-to-end action loss. While the proposed noise prediction loss mitigates this issue, further analysis and comparison with other attack methods in terms of computational efficiency would be beneficial.

**Questions:**

• Can the authors provide a more detailed analysis of the transferability of the generated adversarial perturbations across different environments and robot models?

• How does the performance of DP-Attacker compare to existing adversarial attack methods designed for deep neural networks? Are there specific advantages or disadvantages of using DP-Attacker?

• What specific defensive strategies can be implemented to mitigate the identified vulnerabilities of diffusion policies? Can the authors provide a theoretical analysis or simulation results to demonstrate the effectiveness of these defenses?

• What are the potential limitations of the proposed attack method in terms of computational complexity and scalability to large-scale systems?

**Limitations:**

The authors acknowledge the limitations of their work and discuss the broader societal impacts of their research.

---

> ### Author Rebuttal · Authors · 2024-08-06
>
> We thank the reviewer for the valuable feedback and comments. We are glad that the author finds our work novel and impactful for developing more robust DP systems. Below is our response to some of the questions raised by the reviewer.
>
> > Lack of Defense Strategies: While the paper demonstrates the effectiveness of DP-Attacker, it does not explore or propose any defensive strategies to mitigate the identified vulnerabilities. This omission hinders the development of robust DP systems and leaves open the question of how to protect against such attacks.
> >
> >
> > What specific defensive strategies can be implemented to mitigate the identified vulnerabilities of diffusion policies? Can the authors provide a theoretical analysis or simulation results to demonstrate the effectiveness of these defenses?
> >
>
> We refer the reviewer to **general response 3** on defense strategies.
>
> > Limited Attack Scenarios: The paper focuses on two specific attack scenarios: hacking the camera and attaching adversarial patches. Exploring a broader range of attack scenarios, such as manipulating the robot's sensors or exploiting other weaknesses in the DP system, would provide a more comprehensive understanding of the system's vulnerabilities.
> >
>
> Attacking the robot’s sensors (proprioception) in the DP framework is an interesting direction for future research. However, we did not include this because we believed hackers were more likely to manipulate the image input (especially in physical-world attacks).
>
> > Computational Complexity: The paper mentions the computational complexity of gradient calculations for the end-to-end action loss. While the proposed noise prediction loss mitigates this issue, further analysis and comparison with other attack methods in terms of computational efficiency would be beneficial.
> What are the potential limitations of the proposed attack method in terms of computational complexity and scalability to large-scale systems?
> >
>
> The computational complexity and scalability of our attack are interesting research directions. DP models are often quite small now to satisfy the requirement of fast inference as a visual-motor policy. Our proposed optimization is based on single-step noise prediction loss, which ensures fast gradient calculation.
>
> > Can the authors provide a more detailed analysis of the transferability of the generated adversarial perturbations across different environments and robot models?
> >
>
> We refer the reviewer to **general response 5** on the transferability of our attacks.
>
> > How does the performance of DP-Attacker compare to existing adversarial attack methods designed for deep neural networks? Are there specific advantages or disadvantages of using DP-Attacker?
> >
>
> Our DP-Attacker is the first proposed suite of methods to attack diffusion policy. One can view the end-to-end loss comparison as an existing method. As shown in Table 4 in our paper, E2E loss with DDPM is dramatically more computationally heavy and impossible to use for online attacks. Our proposed DP-Attacker is faster and performs better in terms of degrading the policy performance.

---

> > ### Comment · Reviewer_7kiH · 2024-08-14
> >
> > Thank you for your reply, some of my doubts have been resolved. I maintain my original score.

---

### Official Review · Reviewer_FStp · 2024-07-11

**Soundness:** 3
**Presentation:** 3
**Contribution:** 2
**Rating:** 6
**Confidence:** 4

**Summary:**

The paper proposes an adversarial example construction method targeting Diffusion-based Policies, aiming to create malicious observation inputs to diffusion policies that cause them to fail in generating correct actions, leading to robot errors. The authors present attack methods including both untargeted and targeted attacks. The main idea is to construct adversarial examples by optimizing a malicious denoising loss and updating the image according to the gradient. Experiments demonstrate that the generated adversarial examples can lower the success rate of the target diffusion policy.

**Strengths:**

- This is the first adversarial attack against a diffusion-based visuomotor policies.
- The proposed attack framework is comprehensive, including targeted, untargeted, online, and offline attacks.
- The paper is well-written and easy to understand.

**Weaknesses:**

- The proposed method seems to be a straightforward extension of a previous approach applied to images.
- The threat model is not well-defined.
- The attack method makes strong assumptions on attackers, and its effectiveness in real-world scenarios, especially for patched attacks, is unclear.

**Questions:**

Overall, this paper is interesting and presents the first adversarial attack against diffusion-based visuomotor policies. However, I have the following concerns:

- The proposed method seems to be a straightforward extension of prior work [1], with the main modification being the adjustment of the adversarial loss to accommodate the differences between diffusion policy and standard T2I models.

- Unlike related work that uses adversarial examples to prevent painting imitation, this paper attacks the system directly. Can the authors provide a clear threat model, including a well-defined attacker's goal and capabilities? It would be helpful to provide corresponding scenarios.

- The authors claim that DP-Attacker can generate highly transferable perturbations. How is this demonstrated in the experiments? For example, the authors claim that it is actually the encoder being attacked. The diffusion policy uses ResNet-18 as the encoder [2], can the adversarial examples generated by DP-Attacker transfer to other encoders?

[1] Adversarial example does good: Preventing painting imitation from diffusion models via adversarial examples, 2023

[2] Diffusion Policy: Visuomotor Policy Learning via Action Diffusion, 2023

**Limitations:**

One major limitation is the lack of real-world demonstrations for patched attacks. The authors have acknowledged this in the paper. Additionally, I have some questions regarding the authors' claim about the transferability of the generated adversarial example.

---

> ### Author Rebuttal · Authors · 2024-08-06
>
> We thank the reviewer for the useful valuable comments and feedback! We are glad that the reviewer liked our writing and acknowledges our comprehensive attacks. We would like to address the reviewer’s questions below.
>
> > The proposed method seems to be a straightforward extension of a previous approach applied to images.
> The proposed method seems to be a straightforward extension of prior work [1], with the main modification being the adjustment of the adversarial loss to accommodate the differences between diffusion policy and standard T2I models.
> >
>
> Yes, previous works like [24, 25] propose attacks for LDMs, and [25] further point out that the mechanism of why LDM is vulnerable is because the noises are scaled up by the encoder of the diffused latents.
>
> However, Diffusion Policy is different from LDMs since the attacked item is the condition instead of the diffused latent; as far as we know, no previous works have conducted attacks against the conditional image of diffusion models.
>
> We refer the reviewer to our **general response 1** for more details about the novelty of our method.
>
> > The threat model is not well-defined. Unlike related work that uses adversarial examples to prevent painting imitation, this paper attacks the system directly. Can the authors provide a clear threat model, including a well-defined attacker's goal and capabilities? It would be helpful to provide corresponding scenarios.
> >
>
> We have added a detailed definition of the threat model in general response 4. We will also add this clarification to the final paper.
>
> > The attack method makes strong assumptions on attackers, and its effectiveness in real-world scenarios, especially for patched attacks, is unclear.
> >
>
> We refer the reviewer to **General Response 2** for attacks in the real world.
>
> > The authors claim that DP-Attacker can generate highly transferable perturbations. How is this demonstrated in the experiments? For example, the authors claim that it is actually the encoder being attacked. The diffusion policy uses ResNet-18 as the encoder [2], can the adversarial examples generated by DP-Attacker transfer to other encoders?
> >
>
> That's a good question. We conduct experiments in **general response 5** for the transferability of the attacks across different model structures (CNN and Transformer). For the visual encoder, while there are no pre-trained DP to use, we think there is still some level of transferability according to previous investigations into different model structures.

---

> > ### Comment · Reviewer_FStp · 2024-08-11
> >
> > Thank the authors for their responses. I find the approach of attacking the conditional input instead of the denoising input to be interesting. However, I feel that the overall novelty is still limited, so I did not increase my score.

---

> > > ### Author Response · Authors · 2024-08-12
> > >
> > > Thanks for your reply.  We aim to build comprehensive exploration into the robustness of DP, which turns out to be a promising backbone choice for behavior cloning.
> > >
> > > The difference of inputs is one point，another major difference is in the global attacks, we need to optimize one noise for all frames which have not been done before.
> > >
> > > Overall, we believe our contribution is good to the community and many following works can be inspired. Thanks again for your acknowledgment!

---

### Official Review · Reviewer_fJsu · 2024-07-12

**Soundness:** 3
**Presentation:** 3
**Contribution:** 2
**Rating:** 4
**Confidence:** 4

**Summary:**

This paper presents strategies to attack visual-based diffusion policy networks. The authors investigated two attacking scenarios: hacking the scene camera by adding imperceptible digital perturbations and hacking the scene by attaching small adversarial patches to the environments.

**Strengths:**

1. The paper is very well-written.
2. Authors have shown experimental evidence that diffusion-based policy networks are vulnerable to adversarial attacks in digital and physical-domain settings.

**Weaknesses:**

1. Although authors have demonstrated that diffusion-based policy networks are not robust and susceptible to adversarial perturbations in visual inputs, their attacks are not novel. PGD and patch-based attacks are very common. From this perspective, this work does not contribute substantially.
2. The authors mentioned in the related work section that physical-world attacks are always based on patches. This claim is not true. Many existing physical adversarial attacks are not based on patches. (e.g., Adversarial laser, adversarial shadows, etc.). The
3. The authors mentioned in the limitations section that they have not evaluated this method in real-world settings, so it is hard to assess its practicality.
4. The authors' main contribution is the demonstration that diffusion policy networks are vulnerable to adversarial attacks. Whenever the input is corrupted, it affects the model's output. It is unclear how likely a policy network or robot (in case of physical attacks) is to face these perturbations in real life. It would be great if authors could focus on crafting application-specific attacks with a high chance of occurring in real life.

**Questions:**

1. Is there a reason authors have considered PGD and patch-based attacks? Did they consider crafting realistic and novel attacks?

**Limitations:**

I appreciate that the authors have clearly mentioned their work's limitations. But my biggest concern is the use of already-existing (very common) visual perturbations. I suggest the authors craft attacks that are more realistic, challenging to implement, and have a higher probability of being seen in the real world.

---

> ### Author Rebuttal · Authors · 2024-08-06
>
> We are grateful for the reviewer's comments and valuable feedback. We are delighted that the author liked our writing and is convinced by our algorithms. We appreciate that the reviewer is convinced that diffusion-based policy networks are susceptible to adversarial attacks. Here’s our response to the reviewer’s concerns:
>
> > W1 Although authors have demonstrated that diffusion-based policy networks are not robust and susceptible to adversarial perturbations in visual inputs, their attacks are not novel. PGD and patch-based attacks are very common. From this perspective, this work does not contribute substantially.
> >
>
> The focus of our work is to derive a suite of tools that could successfully hijack diffusion-based imitation learning and reveal that there is still a long barrier to making this algorithm safe for deployment. We have successfully reviewed this by developing fast and efficient online attacks, transferable offline attacks, and physically realizable patch attacks. We list some points in the **general response 1** about the comparison of our work with existing methods and why it is challenging.
>
> > W2 The authors mentioned in the related work section that physical-world attacks are always based on patches. This claim is not true. Many existing physical adversarial attacks are not based on patches. (e.g., Adversarial laser, adversarial shadows, etc.).
> >
>
> Thank you for pointing this out to us. Adversarial patches are widely used in real-world attacks [3, 4, 5], especially in robotics applications [6, 7]. Future work can be done on exploring the robustness of DP against different types of realistic attacks.
>
> Adversarial lasers and shadows are interesting forms of physical attacks. We have found two works that have used them in physical world attacks. [1] focuses on adversarial lasers, and [2], which focuses on adversarial shadows. However, we believe patch-based attacks will be more suitable for conducting physical attacks for **diffusion policy**. [1] and [2] focus on attacking **DNNs for classification**. As mentioned by [1], the effectiveness of these attacks either creates new visual cues that lure DNNs to misclassify or perform as a dominant feature of a set of classes of objects. This logic does not transfer that well to diffusion policy, which is a visual motor policy that uses images as conditions to predict continuous actions.
>
>
>
>
> > W3 The authors mentioned in the limitations section that they have not evaluated this method in real-world settings, so it is hard to assess its practicality.
> >
>
> We refer the reviewer to our **general response 2** for discussion on real-world attacks.
>
> > W4 The authors' main contribution is the demonstration that diffusion policy networks are vulnerable to adversarial attacks. Whenever the input is corrupted, it affects the model's output. It is unclear how likely a policy network or robot (in case of physical attacks) is to face these perturbations in real life. It would be great if authors could focus on crafting application-specific attacks with a high chance of occurring in real life.
> >
>
> To answer the lack of real-world experiments, we refer to the answer for **general response 2**. In this work, we focus more on exposing the security risk of diffusion policy, so we have chosen patched attacks. As answered in W3 we believe other forms of physical attacks to be less effective for attacking DP. We will discuss the limitation that we have not thought of making physical attacks more realizable and will point in the direction of making less conspicuous patches as attacks.
>
> >Q1 Is there a reason authors have considered PGD and patch-based attacks? Did they consider crafting realistic and novel attacks?
> >
>
> Since our attack scenario is white-box, PGD is one of the best tools for gradient-based adversary construction. As we have mentioned in response to W2 and W4, we believe that physical patches are good candidates for attacking diffusion policies.
>
>
> [1] R. Duan et al., “Adversarial Laser Beam: Effective Physical-World Attack to DNNs in a Blink,” Mar. 11, 2021, arXiv: arXiv:2103.06504. doi: 10.48550/arXiv.2103.06504.
>
> [2] Y. Zhong, X. Liu, D. Zhai, J. Jiang, and X. Ji, “Shadows can be Dangerous: Stealthy and Effective Physical-world Adversarial Attack by Natural Phenomenon,” Mar. 22, 2022, arXiv: arXiv:2203.03818. doi: 10.48550/arXiv.2203.03818.
>
> [3] Y. Mirsky, “IPatch: a remote adversarial patch,” Cybersecurity, vol. 6, no. 1, p. 18, May 2023, doi: 10.1186/s42400-023-00145-0.
>
> [4] T. B. Brown, D. Mané, A. Roy, M. Abadi, and J. Gilmer, “Adversarial Patch,” May 16, 2018, arXiv: arXiv:1712.09665. doi: 10.48550/arXiv.1712.09665.
>
> [5] Y.-C.-T. Hu, J.-C. Chen, B.-H. Kung, K.-L. Hua, and D. S. Tan, “Naturalistic Physical Adversarial Patch for Object Detectors,” in 2021 IEEE/CVF International Conference on Computer Vision (ICCV), Montreal, QC, Canada: IEEE, Oct. 2021, pp. 7828–7837. doi: 10.1109/ICCV48922.2021.00775.
>
> [6] A. Tanev, S. Pavlitskaya, J. Sigloch, A. Roennau, R. Dillmann, and J. M. Zollner, “Adversarial Black-Box Attacks on Vision-based Deep Reinforcement Learning Agents,” in 2021 IEEE International Conference on Intelligence and Safety for Robotics (ISR), Tokoname, Japan: IEEE, Mar. 2021, pp. 177–181. doi: 10.1109/ISR50024.2021.9419509.
>
> [7] Y. Jia, C. M. Poskitt, J. Sun, and S. Chattopadhyay, “Physical Adversarial Attack on a Robotic Arm,” IEEE Robot. Autom. Lett., vol. 7, no. 4, pp. 9334–9341, Oct. 2022, doi: 10.1109/LRA.2022.3189783.

---

> > ### Comment · Reviewer_fJsu · 2024-08-12
> >
> > Due to the transformations that happen in the physical world, the perturbations injected in the digital world are sometimes not generalizable in physical settings. Demonstrating the vulnerability of policy networks to adversarial threats is not a significant contribution. For this reason, I will not increase my current score.

---

> > > ### Author Response · Authors · 2024-08-12
> > >
> > > Thanks for your reply
> > >
> > > > Due to the transformations that happen in the physical world, the perturbations injected in the digital world are sometimes not generalizable in physical settings.
> > >
> > >
> > > Our physical patched attacks simulates the perturbations in the real world settings (including transformations and lightenings), which can be done by using render engine. It can highly reflect the performance for real world settings.
> > >
> > > > Demonstrating the vulnerability of policy networks to adversarial threats is not a significant contribution.
> > >
> > >
> > > Diffusion policy is not the same as previous end to end policy network. The structure is very different, and the robustness of it under different kinds of attacks remains unexplored. It is actually not that intuitive why it is vulnerable because previous works find that non-latent diffusion model is **very robust** [a]. Here we contribute the vulnerability to the visual encoder, further work can be conducted to make it more robust by using better vision encoder.
> > >
> > >
> > > [a] Pixel is a barrier: Diffusion Models are more Robust Than we Think

---

### Official Review · Reviewer_7zbu · 2024-07-13

**Soundness:** 3
**Presentation:** 3
**Contribution:** 2
**Rating:** 4
**Confidence:** 3

**Summary:**

This paper studies the adversarial attack to diffusion policy. Two attack scenario settings are introduced. One is to attack the scene camera by adding imperceptible digital perturbations to the visual observation. The other is to attack the scene by adding small adversarial patches to the environment. Experiments show promising results.

**Strengths:**

1. This paper is easy to follow and well-organized.
2. The motivation to attack diffusion policy is interesting.
3. Two hacking settings are studied.

**Weaknesses:**

1. The technical novelty is marginal. The proposed framework directly applies the existing attack method ([25, 24, 52] in the Reference) to diffusion policy in two settings. This paper is more like an experimental report by applying adversarial attack methods to diffusion policy in different settings.
2. The investigation on diffusion policy attack is very shallow. Deeper analysis is needed for improving the contribution.

**Questions:**

1. Is there any specific challenge for applying the existing attack methods to diffusion policy?
2. Technically, what is the technical difference between attacking the conditional images of diffusion policy and those conditional images of diffusion models for other tasks, i.e., anomaly detection?

---

> ### Author Rebuttal · Authors · 2024-08-06
>
> We thank the reviewer for their insightful comments and feedback. Below is our response to address some of the questions raised in the review. We hope that it will address some of your concerns:
>
> > W1 The technical novelty is marginal. The proposed framework directly applies the existing attack method ([25, 24, 52] in the Reference) to diffusion policy in two settings. This paper is more like an experimental report by applying adversarial attack methods to diffusion policy in different settings.
> >
>
> Yes, previous works like [24, 25] propose attacks for LDMs by perturbing the diffused image, and [25] further point out that the mechanism of why LDM is vulnerable is because the noise are scaled up by the encoder of the diffused latents.
>
> However, Diffusion Policy is different from LDMs since the attacked item is the condition instead of the diffused latent; as far as we know, no previous works have conducted attacks against the conditional image of diffusion models.
>
> Also, the diffusion policy is a decision model that includes iterative interaction with the environments; attacking such a diffusion-based system remains an open field before our work.
>
> We refer the reviewer to our **general response 1** for more details about the novelty of our work. Attacking diffusion policy is fundamentally different from attacking LDMs for T2I tasks. Our proposed DP-Attacker successfully dramatically decreases the performance of DP in simulation.
>
> > W2 The investigation on diffusion policy attack is very shallow. Deeper analysis is needed for improving the contribution.
> >
>
> We are the first to investigate adversarial attacks against diffusion policy. We focus on providing a set of effective attack algorithms and verifying their effectiveness. We also provide some insights into why diffusion policy can be easily attacked.
>
> > Q1 Is there any specific challenge for applying the existing attack methods to diffusion policy?
> >
>
> We have mentioned a few challenging aspects of attacking diffusion policy in **general response 1**.
>
> > Q2 Technically, what is the technical difference between attacking the conditional images of diffusion policy and those conditional images of diffusion models for other tasks, i.e., anomaly detection?
> >
>
> This is an interesting question. We found the following paper on using diffusion models for anomaly detection with conditional images [1].  In this work, the image condition is used directly to guide the generation process by modifying the score with a mathematical formula. This differs from the diffusion policy, where the image condition is used as an input of DNN to predict the noise/score.
>
> Nevertheless, attacking diffusion policy is a new task.  DP is a fundamental algorithm for embodied AI, we are the first to evaluate the robustness of it.
>
> [1] A. Mousakhan, T. Brox, and J. Tayyub, “Anomaly Detection with Conditioned Denoising Diffusion Models,” Dec. 03, 2023, arXiv: arXiv:2305.15956. Accessed: Aug. 05, 2024. [Online]. Available: http://arxiv.org/abs/2305.15956

---

> > ### Comment · Reviewer_7zbu · 2024-08-11
> > **Post-rebuttal**
> >
> > Thank the authors for answering my questions. I still believe that this is a borderline paper. I still think that the task studied in this paper is novel while the technical novelty is limited, as it applies the existing  attack techniques including diffusion attack methods [25, 24, 52] to robotic diffusion policy. I am good with the paper to be accepted while I would like keep my original score.

---

> > > ### Author Response · Authors · 2024-08-11
> > >
> > > Thanks for your reply. We are happy to see that we addressed most of the questions.
> > >
> > > For the technique novelty, we listed the challenging points of tasks in the general response. While the specific methods for online global attacks do not differ a lot from attacking DM, the methods we proposed to do online attack is never done before because we need to generate the same perturbations across different frames.
> > >
> > > Overall, we believe our contribution is good to the community and many following works can be inspired.

---

### Official Review · Reviewer_69xg · 2024-07-14

**Soundness:** 3
**Presentation:** 3
**Contribution:** 3
**Rating:** 4
**Confidence:** 4

**Summary:**

The paper analyzes the security vulnerabilities of the diffusion strategy and proposes possible attack scenarios.
A set of algorithms called DP-Attacker is proposed, which can successfully reduce the performance of the diffusion strategy in different adversarial scenarios (including online and offline attacks).

**Strengths:**

This paper proposes a novel adversarial attack framework, specifically targeting diffusion model-based strategies. It not only analyzes the security vulnerabilities of diffusion strategies, but also successfully implements effective attacks on these strategies, demonstrating powerful attack capabilities.

The paper verifies the effectiveness of the attack algorithm through extensive experiments. These experiments not only include attacks on existing diffusion strategy models, but also cover attack scenarios under different settings, such as online and offline attacks, and provide comprehensive experimental results and analysis.

**Weaknesses:**

1. The method used is not specific to the diffusion model.
2. Novelty is limited.

**Questions:**

What is unique about this approach compared to a large number of previous black-box and white-box attacks?

---

> ### Author Rebuttal · Authors · 2024-08-06
>
> We thank the reviewer for the questions raised. Please see our response below. We sincerely ask the reviewer to refer to the general response for possible concerns.
>
> > Q1: The method used is not specific to the diffusion model. Novelty is limited. What is unique about this approach compared to a large number of previous black-box and white-box attacks?
> >
>
> We refer the reviewer to **general response 1** for the novelty of our work. Our algorithm is specifically designed for diffusion policy to ensure online fast adversary generation. To achieve this, we have used a Monte Carlo approximation with the noise prediction loss rather than the end-to-end loss.

---

### Official Review · Reviewer_kPb5 · 2024-07-14

**Soundness:** 4
**Presentation:** 3
**Contribution:** 2
**Rating:** 5
**Confidence:** 3

**Summary:**

Diffusion policy is used to generate the action trajectory from a pure Gaussian noise conditioned on the input images, applied in many applications such as autonomous driving. This paper proposes white-box adversarial attacks against diffusion policy, which aim to generate a target bad action or an untargeted action by attaching global perturbation or patch-based perturbation on the observation image. To this end, they formulate the training loss for the perturbation as minimizing the distance between the generated action and the target bad action (in targeted attack) or maximizing the distance between the generated action and a sampled good solution (in untargeted attack). Empirically, they validate the effectiveness of the proposed adversarial attack on six robotic manipulation tasks.

**Strengths:**

1. They explored adversarial attacks in a new setting, i.e., against diffusion policy.
2. They have good visualizations of experimental results.

**Weaknesses:**

1. As mentioned in the limitation section, there is no intuitive defensive strategy or experimental results in real-world scenarios included in the paper.
2. The proposed attack only considers the white-box setting, i.e., the attacker requires knowledge of all parameters of the diffusion policy. Discussion on black-box setting could also be included in the limitation section.

**Questions:**

1. Threat model could be clarified with more clearness in problem settings. It appears to me it is a white-box attack where the attacker requires knowledge of all parameters of the diffusion policy.
2. The novelty of the proposed adversarial attack remains unclear to me. On the one hand, there are existing works [25, 43] which propose adversarial attacks against latent diffusion models, introducing  the Monte-Carlo-based adversarial loss. On the other hand, the difficulty of computing gradients due to the inherent long-denoising chain of the diffusion policy is solved by [25, 24, 52]. So, it seems that the novelty over these existing works is that this paper is modifying the training loss as the difference between actions. Clarification of the novelties compared with existing works would be appreciated.
3. Is there any other evaluation metric in existing works? It seems that there is only one evaluation metric used (i.e., the drop of the result task completion scores) in the experimental results to demonstrate the effectiveness of the attack. For me, this one single metric would be fine for the untargeted attacks; however, for targeted attacks, it might be better to show the success rate of generating the target bad actions.

**Limitations:**

See weakesses part.

---

> ### Author Rebuttal · Authors · 2024-08-06
>
> We thank the reviewer for the valuable comments. We are glad that the reviewer found our attack scenario novel and liked our visualizations. Below is our response to some of the question raised.
>
> First, we would like to clarify the loss used in our DP-Attacker is not the distance between generated actions. Instead, we use a Monte-Carlo-based estimate with a loss between predicted noise and actual noise. In this way, we were able to significantly speed up the adversary construction process while maintaining good attack abilities. Below is our response to address the weaknesses and questions raised in the review.
>
> > W1 As mentioned in the limitation section, there is no intuitive defensive strategy or experimental results in real-world scenarios included in the paper.
> >
>
> We refer the reviewer to sections **2** and **3** of our **general response**. These are interesting directions for future research.
>
> > W2 The proposed attack only considers the white-box setting, i.e., the attacker requires knowledge of all parameters of the diffusion policy. Discussion on black-box setting could also be included in the limitation section.
> >
>
> We are not considering black-box attacks in this work. We will add a discussion of black-box attacks in the limitation section.
>
> > Q1 Threat model could be clarified with more clearness in problem settings. It appears to me it is a white-box attack where the attacker requires knowledge of all parameters of the diffusion policy.
> >
>
> In **general response 4**, we have clarified the threat model. We will also incorporate this into the final paper.
>
> > Q2 The novelty of the proposed adversarial attack remains unclear to me. On the one hand, there are existing works [25, 43] which propose adversarial attacks against latent diffusion models, introducing the Monte-Carlo-based adversarial loss. On the other hand, the difficulty of computing gradients due to the inherent long-denoising chain of the diffusion policy is solved by [25, 24, 52]. So, it seems that the novelty over these existing works is that this paper is modifying the training loss as the difference between actions. Clarification of the novelties compared with existing works would be appreciated.
> >
>
> We have a detailed subsection addressing this in the **general response 1**.
>
> > Q3 Is there any other evaluation metric in existing works? It seems that there is only one evaluation metric used (i.e., the drop of the result task completion scores) in the experimental results to demonstrate the effectiveness of the attack. For me, this one single metric would be fine for the untargeted attacks; however, for targeted attacks, it might be better to show the success rate of generating the target bad actions.
> >
>
> The success rate is the most direct way for imitation learning to evaluate the effectiveness of a model. Although other continuous reward functions can also be used, the decrease in success rate will be enough to show whether the attacks have largely affect the system.
>
>  The second point for evaluating targeted attacks is interesting. Previously, we only evaluated the targeted attacks qualitatively (see the videos at the bottom of our video website). Here, we have added an experiment that measures the distance between the model's output actions after the attack and target actions. Diffusion policy outputs a sequence of actions for the robot to execute. We use DP-Attacker with different strengths to attack it with some target action sequences and measure the average L2 distance between the output action and targeted action over the length of the action. Using predicted action instead of actual agent position is justified by the fact that the low-level controller will execute these actions without any checking. We report the results in a line graph that shows the distance over the environmental steps during the whole rollout. All attacks have step number $N=50$ and $\alpha=\frac{2\delta}{N}$.
>
> Please see the **newly added PDF Figure 1**. for the results.
>
> In the first scenario, we used DP-Attacker on the PushT (CNN) task to run global online attacks. The target is set to a 2D coordinate around (323.875, 328.75). Note the side length of the PushT task action space is 1024. See the figure in the added pdf. We were able to manipulate the generated action to be within 20 units of the target coordinate with attack strength $\delta=0.06$.
>
> In the second scenario, we used DP-Attacker on the CAN (PH CNN) task to run global online attacks. The end-effector (EE) target is set to a 3D coordinate around (0.1686, 0.1049, 1.0848). The unit is meters. To simplify the metric, we did not set a target for the EE pose or the gripper opening or closing (or else we would need some distance calculating scheme for the 7D output). With attack strength $\delta=0.06$, we could manipulate the generated action within 5 centimeters close to the target position. Both experiments show the effectiveness of our method.
>
> Example attacked frames of different strengths have been added to the pdf as well (please open it with Adobe Acrobat Reader to view the animated frames). Please see **Figure 2~7**.

---

> > ### Comment · Reviewer_kPb5 · 2024-08-10
> >
> > Q1: Thanks for the clarification on the threat model.
> >
> > Q2: I appreciated your clarification on the novelty compared with adversarial attacks on LDMS, especially: 1) "Diffusion policy is a “dynamic” policy network, whereas LDMs are more “static” models in their uses." 2) "the attacked image is the denoising input of the LDM network. However, in diffusion policy, the image is the conditional input of the diffusion model, and the denoising input is the robot's actions."
> >
> > Q3: Thank you for incorporating the distance between the generated action sequence and the target action sequence. I recommend including these findings in the final paper. While qualitative observations confirm successful generation of the target, presenting quantitative results would enable future comparisons and enhance the paper's contribution.

---

> > > ### Author Response · Authors · 2024-08-11
> > > **Thanks for you reply**
> > >
> > > Thanks for your reply. We are happy to see that we have addressed your concerns! We will include the results during the rebuttal into the final version.

---

### Author Rebuttal · Authors · 2024-08-06

We thank the reviewers for their valuable comments and feedback on our paper. We are delighted that all the reviews find our method a novel and effective attack against diffusion policies. We are glad that the **reviewers kPb5 and 69xg** are convinced by our experimental visualizations and results. **Reviewers 7zbu, fJsu, and FStp** found our paper well-written and easy to follow.

In the following section, we would like to address some of the common questions raised by the reviewers:

> 1. Comparison of DP-Attacker with existing methods.

We are the first to present a set of algorithms (DP-Attacker) that can successfully attack diffusion policy, one of the most popular methods for imitation learning in robotics. We focus on exploring the attackability of diffusion policy and devising new adaptive attacks for this special task and special structure, which is an open question. Our method is based on previous adversarial attacks for DNNs and LDMs. However, none of them **can be directly used** to attack diffusion policy effectively since:

1. Diffusion policy is a “dynamic” policy network, whereas LDMs are more “static” models in their uses. The policy network is invoked at a high frequency, requiring fast adversary construction for online settings. Our method successfully achieved this with carefully designed loss and algorithms. We also devised offline attacks that can function across the whole policy rollout.

2. Although methods have been developed for attacking diffusion models (DMs) for image generation [24, 25, 43, 52], the problem differs from how diffusion policy is formulated. In attacking image DMs, which are mostly latent DMs, where the image generation is processed through SDEdit, the attacked image is the denoising input of the LDM network. However, in diffusion policy, the image is the conditional input of the diffusion model, and the denoising input is the robot's actions.

3. Randomness is injected during the denoising process while generating an action in diffusion policy. This might make the policy more robust to adversarial inputs.


> 2. Lack of real world experiments.

First, the digital attacks (e.g. online attacks in our settings) will not be affected by real-world settings, since the attacks are generated to be directly added to the image.

For adversarial physical patches, we try our best to make it as close as the real settings in the real world by evaluating our attack with a "physically" stuck patch in the simulation:

The patch is put into the photo-realistic simulation environment provided by robomimic and robosuite and has been shown to decrease the performance of diffusion policy successfully. Real-world data is more complex and is challenging to learn for diffusion policy. We conducted our evaluation in simulation to show the effectiveness of our attack method better. Models trained on real-world data may also be more sensitive to adversarial perturbations. Thus, we believe our proposed DP-Attacker will work in the real world as well.

For further research, we will try to apply it to the real world.

> 3. What could be the defense strategy?

While the defense of DP-Attacker is not the focus of this work, we provide some possible defenses:

- (1) purify the observation using diffusion-based purification methods

- (2) apply adversarial training to increase the robustness of the image encoder

> 4. Could you clarify the threat model?

In our work, we only consider white-box attacks on diffusion policy (DP) in which the attacker can access the model, its parameters, and the data used to train it. The attacker's goal is to decrease the performance of DP (task score or success rates). In targeted attacks, we also wish to be able to control the model’s generated trajectory. Two attacking scenarios are explored (Fig. 1 in the manuscript). In the first scenario, the hacker is allowed to modify every pixel of the image with some budget $\delta$. Afterward, the modified image is used for DP inference and rollout. We develop two types of perturbations; one is calculated online and is generated per inference. The second type is pre-generated and used throughout the rollout. In the second scenario, the attacker puts a pre-generated colored patch in the camera's view. The patch undergoes a physical process of reflection and camera imaging before being used for inference in the model.

> 5. What is the transferability of your attacks?

Originally, we used transferability to indicate that our devised offline attacks (global or patched) can function to disrupt the performance of the DP across the rollout of the policy where the input image is consistently changing. However, since the reviewers are also interested in the transferability of our attacks across models, we conducted the following experiment. We believe it is more reasonable to test the transferability of offline attacks rather than online attacks because online attacks are often very specific to the model with white-box access and will not transfer well to other networks.

We first tested the transferability of offline global attacks. We used DP-attacker to generate untargeted global offline attacks ($\delta=0.03$) on two checkpoints (CAN-MH-CNN and CAN-MH-TF). Then, we evaluated these models using the two generated adversarial perturbations, and the success rate is listed in Table 1 of the attached PDF.

| Runner Model\Attacked Model | CNN | TF | Original SR |
| --- | --- | --- | --- |
| CNN | 0.34 | 0.78 | 0.98 |
| TF | 0.32 | 0.46 | 0.92 |

We also tested the transferability of patched attacks. We used DP-Attacker to generate adversarial patches on two checkpoints (CAN-PH-CNN and CAN-PH-TF). Then, we evaluated these models using the two generated adversarial perturbations, and the success rate is listed in Table 2 of the attached PDF.

| Runner Model\Attacked Model | CNN | TF | Original SR |
| --- | --- | --- | --- |
| CNN | 0.16 | 0.54 | 0.98 |
| TF | 0.42 | 0.44 | 0.92 |

---

### Decision · Program_Chairs · 2024-09-25

**Decision:**

Accept (poster)

**Comment:**

The paper proposes a new attack against "diffusion policy" (DP) models, whose underlying goal is to craft tiny perturbations that can mislead these models so that the device receiving the model's output takes a wrong action.

Six reviewers reviewed the paper. Three reviewers provided positive scores (7, 6, 5), while three reviewers are more on the negative side (proposing a 4). No flaws were identified during the reviewing phase, indicating that the results are likely to be correct and representative of real scenarios. The reviewers all acknowledged that the paper tackles a relatively underinvestigated aspect (from a security standpoint) of diffusion-based policies, which have plenty of real-world relevance. Some ethical concerns were identified, but have been deemed to be addressable by the Ethical reviewers. However, the major concerns identified were due to a limited novelty of the proposed attack from a methodological perspective.

Indeed, the proposed attack seems to borrow its fundamental principles from prior work, and the results appear to be also somewhat expected, especially given that the paper envisions a "white-box" attacker.

While this is a noteworthy concern (as white-box attacks are tough to stage in the real world), I believe that the paper should be accepted. This is due to various factors, such as:

* the paper is well written and presented
* the results show that the attack is effective
* the paper does consider a "novel" setting (from an adversarial ML perspective)
* the paper carries out an experiment envisioning a "physical" adversarial patch which aligns with a real-world environment
* the interactive discussion shed more light on two important properties of the attacks: their transferability as well as potential defenses

Given the above, the paper is likely to create interest in the NeurIPS audience (due to the widespread usage of DP) and while not providing groundbreaking results, it may represent a stepping stone for future research (which can, e.g., consider more constrained threat models; or develop ad-hoc countermeasures).